# NSNet: A General Neural Probabilistic Framework for Satisfiability Problems

**Zhaoyu Li**[1,2]**, Xujie Si**[1,2,3]
[1]McGill University, [2]Mila – Quebec AI Institute, [3]CIFAR AI Research Chair
`{zli199, xsi}@cs.mcgill.ca`

## Abstract

We present the Neural Satisfiability Network (NSNet), a general neural framework that models satisfiability problems as probabilistic inference and meanwhile exhibits proper explainability. Inspired by the Belief Propagation (BP), NSNet uses a novel graph neural network (GNN) to parameterize BP in the latent space, where its hidden representations maintain the same probabilistic interpretation as BP. NSNet can be flexibly configured to solve both SAT and #SAT problems by applying different learning objectives. For SAT, instead of directly predicting a satisfying assignment, NSNet performs marginal inference among all satisfying solutions, which we empirically find is more feasible for neural networks to learn. With the estimated marginals, a satisfying assignment can be efficiently generated by rounding and executing a stochastic local search. For #SAT, NSNet performs approximate model counting by learning the Bethe approximation of the partition function. Our evaluations show that NSNet achieves competitive results in terms of inference accuracy and time efficiency on multiple SAT and #SAT datasets [1].

## 1 Introduction

The Boolean Satisfiability Problem (SAT) and the Sharp Satisfiability Problem (#SAT, or model counting) are fundamental challenges for computer science, with numerous applications including software verification [12, 19], hardware design [28, 36], and planning [14, 15]. Although modern SAT and #SAT solvers have achieved practical success in these domains, the performance of satisfiability solvers heavily relies on custom search heuristics [8]. However, designing good heuristics is highly non-trivial and time-consuming. With the superior learning ability of neural networks, recent research on satisfiability problems has migrated from hand-engineered methods toward data-driven ones.

Various neural methods have been proposed for satisfiability problems [18]. A line of research aims at building standalone neural solvers, which directly predicts the satisfiability [35, 22, 10] or a satisfying assignment [2, 3, 31] of a given instance. Other approaches integrate the neural modules into classic solvers, improving the branching heuristics in SAT [34] or #SAT [42] solvers. Yet, despite recent progress, the interpretation of these neural networks still remains a mystery: *why and how can neural networks learn to tackle satisfiability problems?* Answering this question is crucial for utilizing neural networks on satisfiability problems, especially for designing better neural architectures.

This paper addresses the above question by developing an explainable neural framework that can be interpreted using graphical models. Specifically, we propose **N**eural **S**atisfiability **Net**work (NSNet), a novel graph neural network (GNN) framework taking the formulation of solving satisfiability problems as probabilistic inference. Inspired by the inference algorithm Belief Propagation (BP), NSNet adopts a new graph encoding for propositional formulas and employs a novel message passing mechanism that subsumes the BP's updating rules in the latent space. In such a manner, NSNet serves

---

[1]Code is available at `https://github.com/zhaoyu-li/NSNet`.

36th Conference on Neural Information Processing Systems (NeurIPS 2022).

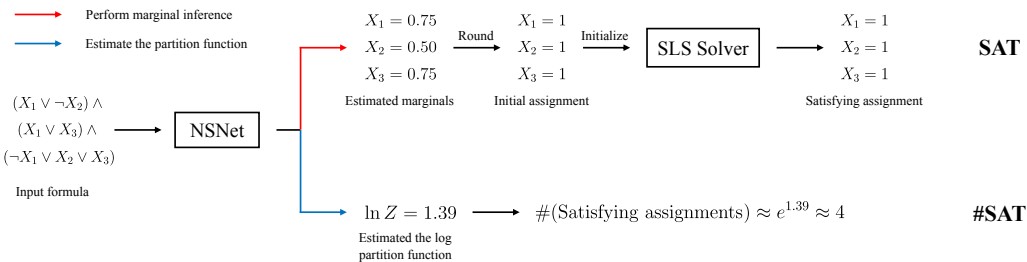

Figure 1: Overview of our pipeline for solving SAT and #SAT problems. For SAT, we apply NSNet to perform marginal inference and use the estimated marginals to guide the initialization of a SLS solver. For #SAT, NSNet itself can serve as an approximate solver by estimating the partition function.

as a neural extension of BP and allows the inference to learn from data. Like BP, NSNet can estimate both marginals and the partition function of graphical models. This key observation makes NSNet an effective unified solution to both SAT and #SAT problems (Figure 1).

Existing neural SAT solvers [2, 3] aim to predict a *single* satisfying assignment for a satisfiable formula. However, there can be multiple satisfying solutions, making it unclear which particular solution should be generated. Instead of directly predicting a solution, NSNet performs *marginal inference* in the solution space of a SAT problem, estimating the assignment distribution of each variable among *all* satisfying assignments. Although NSNet is not directly trained to solve a SAT problem, its estimated marginals can be used to quickly generate a satisfying assignment. One simple way is to round the estimated marginals to an initial assignment and then perform the stochastic local search (SLS) on it. Our experimental evaluations on the synthetic datasets with three different distributions show that NSNet's initial assignments can not only solve much more instances than both BP and the neural baseline but also improve the state-of-the-art SLS solver to find a satisfying assignment with fewer flips.

To solve #SAT, we formulate it as the *partition function estimation* of graphical models. NSNet learns the Bethe approximation of the partition function and thus acts as an approximate #SAT solver. Experiments on the BIRD and SATLIB benchmarks illustrate that our method notably surpasses both BP and the recent neural approach. Compared with the state-of-the-art approximate #SAT solvers, NSNet can estimate the solutions with more than three orders of magnitude speedup, while still achieving competitive precision.

## 2 Preliminaries

**Satisfiability Problems.** In propositional logic, a Boolean formula consists of Boolean variables and logical operators such as negations ($\neg$), conjunctions ($\wedge$), and disjunctions ($\vee$). It is typical to represent Boolean formulas in conjunctive normal form (CNF), expressed as a conjunction of clauses, each of which is a disjunction of literals (a variable or its negation). Given a CNF formula, the SAT problem asks whether there exists an assignment to its variables that can satisfy the formula, whereas the goal for #SAT is to count the number of all satisfying solutions.

**Factor Graphs.** A factor graph is a bipartite graph with a set of variable nodes connected to a set of factor nodes, where each factor node expresses the dependencies among the variables it is connected. Let $p$ denote a distribution defined over $n$ discrete random variables and $x_i$ be a possible assignment of the $i$-th variable $X_i$. Then we may define the joint probability $p(\boldsymbol{x})$ of an assignment $\boldsymbol{x} = \{x_1, x_2, \ldots, x_n\}$ using $m$ factors $\{f_1, f_2, \ldots, f_m\}$ as:

$$p(\boldsymbol{x}) = \frac{1}{Z} \prod_{a=1}^{m} f_a(\boldsymbol{x}_a), \quad Z = \sum_{\boldsymbol{x}} \left( \prod_{a=1}^{m} f_a(\boldsymbol{x}_a) \right), \tag{1}$$

where each factor $f_a$ takes an assignment of its associated variables $\boldsymbol{x}_a \subseteq \boldsymbol{x}$ as input, $Z$ is the normalization constant, which is also known as the partition function.

In the case of satisfiability problems, each Boolean variable and clause in a CNF formula correspond to a variable and a factor node in a factor graph respectively. Given a factor $f_a$ with its associated

clause $a$, the input $\boldsymbol{x}_a$ represents a possible assignment for the variables appears in clause $a$, and $f_a(\boldsymbol{x}_a)$ takes value 1 if $\boldsymbol{x}_a$ satisfies the clause and 0 otherwise. By taking all factors into account, we derive that $p(\boldsymbol{x}) = 1/Z$ if and only if the assignment $\boldsymbol{x}$ satisfies the entire CNF formula, where the partition function $Z$ indicates the number of all satisfying assignments. In other words, we can consider $p(\boldsymbol{x})$ as a probability measure on the space of all assignments that have a uniform distribution for all satisfying assignments and zero probability for unsatisfying ones.

**Belief Propagation.** BP is a variational inference algorithm to estimate the marginal distributions of variables or the partition function $Z$ in a factor graph. It performs iterative message passing between neighboring variable and factor nodes. Specifically, at the $k$-th iteration, the variable to factor messages $m_{i \to a}^{(k)}(x_i)$ and the factor to variable messages $m_{a \to i}^{(k)}(x_i)$ are computed as following:

$$m_{i \to a}^{(k)}(x_i) = \prod_{c \in \mathcal{N}(i) \setminus a} m_{c \to i}^{(k-1)}(x_i), \quad m_{a \to i}^{(k)}(x_i) = \sum_{\boldsymbol{x}_a \setminus x_i} f_a(\boldsymbol{x}_a) \prod_{j \in \mathcal{N}(a) \setminus i} m_{j \to a}^{(k)}(x_j), \quad (2)$$

where $\mathcal{N}(i)$ and $\mathcal{N}(a)$ denotes the neighbor nodes of node $i$ and node $a$ respectively. BP iteratively updates these messages until convergence or reaching a maximum number of iterations $T$. After this process, the variable beliefs $b_i(x_i)$ and the factor beliefs $b_a(\boldsymbol{x_a})$ can be computed to estimate the marginal distributions over the variables and the factors respectively:

$$b_i(x_i) \propto \prod_{a \in \mathcal{N}(i)} m_{a \to i}^{(T)}(x_i), \quad b_a(\boldsymbol{x}_a) \propto f_a(\boldsymbol{x}_a) \prod_{j \in \mathcal{N}(a)} m_{j \to a}^{(T)}(x_j). \quad (3)$$

Given the variable beliefs and the factor beliefs, BP can calculate a variational approximation of the partition function $Z$. Such an approximation is derived as the Bethe free energy $\mathcal{F} = -\ln Z$ in statistical physics [7], which is defined as:

$$\mathcal{F} = \sum_{a=1}^{m} \sum_{\boldsymbol{x}_a} b_a(\boldsymbol{x}_a) \ln \frac{b_a(\boldsymbol{x}_a)}{f_a(\boldsymbol{x}_a)} - \sum_{i=1}^{n} (|\mathcal{N}(i)| - 1) \sum_{x_i} b_i(x_i) \ln b_i(x_i). \quad (4)$$

## 3 Methodology

In this section, we first rewrite standard BP to suit satisfiability problems with a probabilistic interpretation. Based on such a formulation, we introduce the graph encoding and message passing scheme of NSNet, which stands as a neural version of BP. By formulating SAT and #SAT as probabilistic inference tasks, we show how to utilize NSNet to solve these two problems.

### 3.1 Rewriting Belief Propagation for Satisfiability Problems

For numerical stability, BP can be performed in log space, and its messages can be normalized at every iteration. Note each factor $f_a$ can only take value 1 or 0 for satisfiability problems, thus we can remove the factor term in Equation 2 and rewrite it as:

$$m_{i \to a}^{(k)}(x_i) = -z_{i \to a}^{(k)} + \sum_{c \in \mathcal{N}(i) \setminus a} m_{c \to i}^{(k-1)}(x_i), \quad m_{a \to i}^{(k)}(x_i) = \operatorname*{LSE}_{\boldsymbol{x}_a^* \setminus x_i} \left( \sum_{j \in \mathcal{N}(a) \setminus i} m_{j \to a}^{(k)}(x_j) \right). \quad (5)$$

LSE is the shorthand for the log-sum-exp function, and $z_{i \to a}^{(k)}$ is the normalization constant for the variable to factor messages at each iteration. The subscript $\boldsymbol{x}_a^* \setminus x_i$ means to fix the value $x_i$ for the variable $X_i$ and enumerate other *satisfying* variable assignments in clause $a$. Note the normalization term $z_{i \to a}^{(k)}$ ensures that $\sum_{x_i \in \{0,1\}} \exp(m_{i \to a}^{(k)}(x_i)) = 1$, while there is no such a normalization for $m_{a \to i}^{(k)}(x_i)$. From the perspective of graphical models, there is a probabilistic interpretation for the messages in Equation 5 [9]: the variable to clause message $m_{i \to a}(x_i)$ can be interpreted as the log probability that $X_i$ takes value $x_i$ when clause $a$ is *not* considered and the clause to variable message $m_{a \to i}(x_i)$ represents as the log probability that clause $a$ is satisfied when $X_i$ takes the value of $x_i$. With this interpretation, BP performs message passing to approximate these probabilities iteratively.

However, BP often makes inaccurate estimates or fails to converge for satisfiability problems due to complex logical structures. To improve the inference of BP on satisfiability problems, we propose Neural Satisfiability Network (NSNet), a novel GNN framework that extends BP in the latent space, allowing the inference algorithm to learn from data.

## 3.2 NSNet Framework

**Graph Representation.** Inspired by the procedure of BP, we propose a new graph representation for CNF formulas. As shown in Figure 2, we represent a formula as an undirected bipartite graph with one type of node for each variable assignment and another type of node for each clause. An assignment node and a clause node are connected if the associated variable appears in the clause. The edges between them have two different types based on whether the clause can be satisfied using this variable assignment. Note that our graph representation can exactly express each message in BP: the direct edges from an assignment node to a clause node represent the messages $m_{i\to a}(x_i)$ and the edges from the other direction represent the messages $m_{a\to i}(x_i)$ in Equation 5.

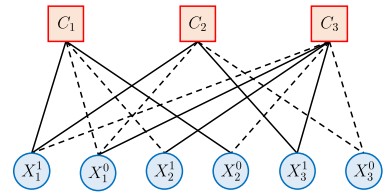

Figure 2: Our graph representation of the CNF formula $(X_1 \lor \neg X_2) \land (X_1 \lor X_3) \land (\neg X_1 \lor X_2 \lor X_3)$. $X_i^1$ and $X_i^0$ denotes variable $X_i$ takes values 1 and 0 respectively. The solid/dashed line indicates that the variable assignment satisfies/dissatisfies the associated clause.

**Message Passing Scheme.** On top of our graph representation, we aim to design a learnable message passing scheme that parameterizes BP with neural modules. Formally, we define an embedding on every *directed* edge in the vector space $\mathbb{R}^d$: the variable assignment to clause embeddings $\boldsymbol{m}_{i\to a}(x_i)$ and the clause to variable assignment embeddings $\boldsymbol{m}_{a\to i}(x_i)$. Initially, we set all the embeddings of $\boldsymbol{m}_{i\to a}(x_i)$ and $\boldsymbol{m}_{a\to i}(x_i)$ to two learnable vectors $\boldsymbol{h}_1$ and $\boldsymbol{h}_2$ respectively. At the $k$-th iteration, these hidden representations are updated as:

$$\tilde{\boldsymbol{m}}_{i\to a}^{(k)}(x_i) = \mathcal{A}_1\left(\sum_{c\in\mathcal{N}(i)\backslash a}\boldsymbol{m}_{c\to i}^{(k-1)}(x_i)\right), \; \boldsymbol{m}_{i\to a}^{(k)}(x_i) = \mathcal{A}_2\left(\tilde{\boldsymbol{m}}_{i\to a}^{(k)}(x_i), \tilde{\boldsymbol{m}}_{i\to a}^{(k)}(1-x_i)\right), \quad (6)$$

$$\boldsymbol{m}_{a\to i}^{(k)}(x_i) = \mathcal{A}_3\left(\underset{\boldsymbol{x}_a^*\backslash x_i}{\mathrm{LSE}}\left(\sum_{j\in\mathcal{N}(a)\backslash i}\boldsymbol{m}_{j\to a}^{(k)}(x_j)\right)\right), \quad (7)$$

where $\mathcal{A}_1, \mathcal{A}_2, \mathcal{A}_3$ are all neural networks, which are parameterized by MLPs in our implementation. At each iteration, we adopt the aggregation operators for updating $\tilde{\boldsymbol{m}}_{i\to a}^{(k)}(x_i)$ and $\boldsymbol{m}_{a\to i}^{(k)}(x_i)$ the same as BP in Equation 5, using summation for $\tilde{\boldsymbol{m}}_{i\to a}^{(k)}(x_i)$ and LSE operation for $\boldsymbol{m}_{a\to i}^{(k)}(x_i)$. The subscript $\boldsymbol{x}_a^* \backslash x_i$ enumerates a set of assignment to clause embeddings connected to clause $a$ except for the assignment $x_i$ and ensures that at least one embedding on a 'satisfying' edge is included during the enumeration. The edge embedding $\boldsymbol{m}_{i\to a}^{(k)}(x_i)$ is updated by incorporating both $\tilde{\boldsymbol{m}}_{i\to a}^{(k)}(x_i)$ and its flip value's $\tilde{\boldsymbol{m}}_{i\to a}^{(k)}(1-x_i)$. With the same aggregation strategy as BP, our framework can be viewed as a generalized BP algorithm that maintains the same probabilistic interpretation of BP and encodes these probabilities in the latent space.

**Proposition 1.** *Our message passing scheme subsumes the BP algorithm, where BP is a special case of our message passing.*

If we set the feature dimension $d$ to 1, and let the neural networks $\mathcal{A}_1$ and $\mathcal{A}_3$ be the identity function, $\mathcal{A}_2$ be the normalization function $\mathcal{A}_2(a,b) = a - \log(\exp(a) + \exp(b))$, then the updating rules in Equation 6- 7 is equivalent to Equation 5. Thus we can consider BP as a non-parameterized GNN and NSNet as a neural version of BP that generalizes it in the vector space with learnable parameters.

In contrast to existing GNN architectures for satisfiability problems [35, 22, 10, 2, 31], NSNet defines latent representations on *edges* and performs message passing between neighboring *edges* rather than *nodes*. Such design also enables NSNet to enforce the permutation invariance and the negation equivariance of CNF formulas [35]:

**Proposition 2.** *For any parameterization of $\boldsymbol{h}_1$, $\boldsymbol{h}_2$, $\mathcal{A}_1$, $\mathcal{A}_2$ and $\mathcal{A}_3$, the embeddings of NSNet is either invariant or equivariant to the following operations: (1) permute any variables. (2) permute any clauses. (3) permute any variables within a clause. (4) negating every literal of a given variable.*

Due to the space limit, we leave the explanation and proof of Proposition 2 in Appendix A.1.

### 3.3 SAT Solving

We first show how NSNet can be used to solve the SAT problem. Although any satisfying assignment is a valid solution to the SAT problem, we need to consider the following question carefully:

**Question 1.** *If a neural network can predict a satisfying assignment, how and why does the neural network produce one particular solution instead of other feasible ones?*

Existing neural methods [22, 10, 2, 3] employ neural networks to produce a value $v \in [0, 1]$ for each variable, where this value is regarded as a soft assignment and rounded to 1 or 0 as the actual assignment. Unfortunately, these approaches fail to answer Question 1: they are trained to generate a specific satisfying solution without considering other possible ones. As a result, neural networks are biased towards specific solutions during training and uncertain about which one to predict during testing, making their generalization ability poor.

To address this question, we leverage NSNet to perform *marginal inference*, i.e., computing the marginal distribution of each variable among all satisfying assignments. In other words, instead of solving a SAT problem directly, we aim to estimate the fraction of each variable that takes 1 or 0 in the entire solution space. Note the marginal for each variable takes all feasible solutions into account and is unique, which is more stable and interpretable to be reasoned by the neural networks. Similar to Equation 3 used by BP to compute variable beliefs, NSNet estimates each marginal value $b_i(x_i)$ by aggregating the clause to variable assignment messages through a MLP and a softmax function:

$$\tilde{b}_i(x_i) = \text{MLP}\left(\sum_{a \in \mathcal{N}(i)} m_{a \to i}^{(T)}(x_i)\right), \quad [b_i(1), b_i(0)] = \text{softmax}\left(\left[\tilde{b}_i(1), \tilde{b}_i(0)\right]\right). \tag{8}$$

To train NSNet to perform marginal inference accurately, we minimize the Kullback-Leibler (KL) divergence loss between the estimated marginal distributions and the ground truth. We use an efficient ALLSAT solver to enumerate all the satisfying assignments and take the average of them to compute the true marginals.

Now we consider how to generate a satisfying assignment after obtaining the estimated marginals. Note that the marginal value $b_i(x_i)$ for the variable $X_i$ represents the probability $X_i$ takes $x_i$ among all satisfying solutions, $X_i$ favors taking the value of 1 to satisfy a formula if $b_i(1) > 0.5$ and vice versa. So one simple way is to round the marginal $b_i(1)$ for each variable to a value $x_i = \lfloor b_i(1) + 0.5 \rfloor$ to produce an assignment. However, when each variable takes its favoring value independently, such an assignment may *not* satisfy the formula even if the estimated marginals are perfect. Therefore, we also perform the local search on this assignment until finding a satisfying one. In practice, we integrate NSNet with a SLS solver by modifying the solver's initialization under the guidance of our estimated marginals. Such a combination is easy to implement and has a modest overhead. It should be stressed that marginals are important in SAT solving as they represent the underlying distribution of individual variables. Besides using local search, one can adopt other methods such as decimation to generate a satisfying solution. We leave the discussion of other approaches in Appendix A.2.

### 3.4 Model Counting

We also apply NSNet to the #SAT problem. Exact model counting is a well-known #P-Complete problem [43] and is almost infeasible without a search procedure. Due to its inherent complexity, a line of research focuses on studying approximate model counting [39, 44, 11, 37, 1, 24], where the goal is to count the number of satisfying assignments *with certain tolerance* at a lower computational cost. Similar to BP, we leverage NSNet to perform approximate model counting by learning the Bethe approximation of the partition function $Z$. Specifically, besides learning the variable beliefs $b_i(x_i)$ using Equation 8, we learn the factor beliefs $b_a(\boldsymbol{x_a})$ defined in Equation 3 by:

$$\tilde{b}_a(\boldsymbol{x_a}) = \text{MLP}\left(\sum_{j \in \mathcal{N}(a)} m_{j \to a}^{(T)}(x_j)\right), \quad b_a(\boldsymbol{x_a}) = \tilde{b}_a(\boldsymbol{x_a}) - \underset{\boldsymbol{x_a}}{\text{LSE}}\left(\tilde{b}_a(\boldsymbol{x_a})\right). \tag{9}$$

Then using Equation 4, the log number of all satisfying assignments can be estimated as:

$$\ln Z = -\sum_{a=1}^{m} \sum_{\boldsymbol{x_a}} b_a(\boldsymbol{x_a}) \ln b_a(\boldsymbol{x_a}) + \sum_{i=1}^{n} (|\mathcal{N}(i)| - 1) \sum_{x_i} b_i(x_i) \ln b_i(x_i). \tag{10}$$

NSNet subsumes the standard Bethe approximation with learnable estimations of variable and factor beliefs, making it a standalone approximate #SAT solver. Unlike SAT which may have multiple satisfying assignments, the solution to a #SAT problem is a unique value. We collect the ground truth value using a well-engineered #SAT solver and train NSNet by minimizing the mean square error between our approximation and the ground truth.

# 4 Experiments

We present the evaluation results in this section. In all experiments, we set the feature dimension $d = 64$ and message passing iteration $T = 10$ for training. All MLPs have 3 hidden layers with 64 units each and use ReLU as the activation function. We ran all experiments on a server with a single NVIDIA A100 GPU and 8 CPU cores. See Appendix B.1 for more implementation details.

## 4.1 SAT Solving

**Experiment Setup.** Following the experiment settings in recent works [35, 2, 3, 31], we experiment using three synthetic SAT generators: SR [35], 3-SAT, and Community Attachment (CA) [17]. The original SR distribution is to generate a set of SAT/UNSAT pairs with only one different literal between the instances in each pair. In our setting, we perform marginal inference on satisfiable instances, so the unsatisfiable ones are discarded. For 3-SAT, we produce random 3-SAT satisfiable instances using CNFgen [26] at the region of the phase transition [13]. CA is the pseudo-industrial generator that produces instances that mimic real-world problems with similar structures. For each distribution, we generate 50k satisfiable formulas with 10 to 40 variables and split them into training/validation/testing sets following 60%/20%/20% proportions. To measure the generalizability of our approach, we further produce 10k larger formulas with the number of variables ranging from 40 to 200. The state-of-the-art ALLSAT solver bdd_minisat_all [41] is used to enumerate all satisfying assignments to obtain the ground truth marginals. More details are in Appendix B.2.1.

**Evaluation & Baselines.** We evaluate our approach for SAT solving in two metrics. The first one is the solving accuracy of the initial assignment obtained from the estimated marginals. We compare NSNet against BP and the neural baseline NeuroSAT [35]. NeuroSAT is the seminal work that designs a literal-clause graph representation for a CNF formula and encodes the structure using a similar gated graph neural network (GGNN) [27]. Such a representation and neural network design are widely used in the following works [22, 10, 34, 47]. We train and evaluate NeuroSAT the same way as NSNet. For these two neural networks, we also examine the effectiveness of our training method against the assignment-supervised training strategy [47, 10, 22], which minimizes the binary cross-entropy loss of the predicted soft assignment and a certain satisfying solution. Glucose4 [5] is used to obtain such a ground truth in this setting. Besides using $T = 10$ message passing iterations, we further test the convergence of all methods by running for more iterations of message passing.

The second metric is the accuracy of a SLS solver with different initial assignments. For this metric, we experiment only on the large testing problems and choose the state-of-the-art pure SLS solver Sparrow [6] as our base solver. We compare our hybrid solver against the Sparrow solver with different initialization strategies, which include the default initialization (random), BP, and NeuroSAT. All SLS solvers are allowed to generate up to 100 assignments for each formula to ensure a fair comparison. Mean and standard deviations are calculated across 10 runs.

**Main Results.** Table 1 shows our initial assignment results on the synthetic datasets using the number of iterations $T = 10$. We can notice that NSNet with marginal supervision obtains the best performance across all distributions. Although our training objective is not for predicting a single satisfying solution, simply rounding the estimated marginals makes both NeuroSAT and NSNet solve much more instances than that trained with the assignment supervision. It is consistent with our hypothesis that fitting on a specific satisfying assignment without considering other possible ones would confuse the neural networks and thus lead to poor generalization. Note that NSNet is only trained on small instances, it is able to generalize to larger ones and outperform BP. Meanwhile, NSNet's performance exceeds NeuroSAT's by a large gap, which also demonstrates the advantage of our graph representation and neural architecture.

Table 1: Solving accuracy (%) of the initial assignments on the synthetic datasets.

| Supervision | Method | Same Distribution | | | | Larger Distribution | | | |
|---|---|---|---|---|---|---|---|---|---|
| | | SR | 3-SAT | CA | Total | SR | 3-SAT | CA | Total |
| N/A | BP | 49.65 | 51.43 | 36.45 | 45.84 | 5.97 | 7.18 | 6.59 | 6.58 |
| Assignment | NeuroSAT | 44.16 | 43.74 | 35.37 | 41.09 | 1.60 | 2.52 | 1.64 | 1.92 |
| | NSNet | 39.62 | 57.63 | 47.20 | 48.15 | 3.37 | 8.13 | 3.61 | 5.03 |
| Marginal | NeuroSAT | 47.77 | 48.60 | 50.97 | 49.11 | 1.99 | 3.18 | 5.61 | 3.59 |
| | NSNet | **63.16** | **63.52** | **56.30** | **60.99** | **9.13** | **12.07** | **8.08** | **9.76** |

Tabel 2 illustrates the averaging solving accuracy with different numbers of iterations on both same-size and larger testing instances. Generally, most approaches can solve more problems with increased iterations of message passing. However, we can observe that the performance of BP hits a saturation at 100 iterations and decreases a lot after then. We conjecture this is because BP's beliefs oscillate a lot with too many iterations, making it fail to converge. Instead, NSNet achieves higher accuracy as the iteration increases and outperforms both BP and NeuroSAT consistently.

Table 2: Solving accuracy (%) of the initial assignments with the increased number of iterations $T$ on the synthetic datasets. $T_{\text{best}}$ stands for the number of iterations achieving the best accuracy.

| Supervision | Method | Number of Iterations ($T$) | | | | | | |
|---|---|---|---|---|---|---|---|---|
| | | 10 | 20 | 50 | 100 | 200 | 500 | $T_{\text{best}}$ |
| N/A | BP | 26.21 | 30.49 | 32.90 | 34.34 | 30.95 | 21.29 | 34.34 |
| Assignment | NeuroSAT | 21.51 | 25.66 | 24.14 | 23.10 | 22.15 | 22.43 | 25.66 |
| | NSNet | 26.59 | 20.75 | 21.18 | 21.40 | 21.59 | 21.78 | 26.59 |
| Marginal | NeuroSAT | 26.35 | 30.89 | 28.41 | 31.10 | 33.35 | 35.21 | 35.21 |
| | NSNet | **35.38** | **40.06** | **41.43** | **41.88** | **42.66** | **42.97** | **42.97** |

In Table 3, we present the performance of SLS solvers with different initialization methods. To reduce the extra overhead of querying models, we perform $T = 10$ iterations of message passing for BP, NeuroSAT, and NSNet. By rounding estimated marginals as initial assignments, BP-Sparrow, NeuroSAT-Sparrow, and NSNet-Sparrow significantly outperform the vanilla Sparrow on all distributions. This fact suggests that marginals are helpful to guide a local search procedure and enhance the SLS solver to find a satisfying assignment efficiently. Among all initialization approaches, NSNet still performs the best. More experimental results for SLS solvers can be found in Appendix B.2.2.

Table 3: Solving accuracy (%) for Sparrow with different initializations on the synthetic datasets.

| Method | Larger Distribution | | | |
|---|---|---|---|---|
| | SR | 3-SAT | CA | Total |
| Sparrow | $8.77 \pm 0.15$ | $11.48 \pm 0.26$ | $54.25 \pm 0.23$ | $24.83 \pm 0.08$ |
| BP-Sparrow | $27.76 \pm 0.20$ | $35.30 \pm 0.31$ | $84.89 \pm 0.19$ | $49.32 \pm 0.11$ |
| NeuroSAT-Sparrow | $22.04 \pm 0.30$ | $29.03 \pm 0.30$ | $83.64 \pm 0.22$ | $44.90 \pm 0.18$ |
| NSNet-Sparrow | $\mathbf{29.66 \pm 0.15}$ | $\mathbf{37.24 \pm 0.18}$ | $\mathbf{86.13 \pm 0.21}$ | $\mathbf{51.01 \pm 0.11}$ |

## 4.2 Model Counting

**Experiment Setup.** We first follow the experiment settings in recent work BPNN [24]. Specifically, we run experiments using the same subset of BIRD benchmark [37], which contains eight categories arising from DQMR networks, grid networks, bit-blasted versions of SMTLIB benchmarks, and ISCAS89 combinatorial circuits. Each category has 20 to 150 CNF formulas, which we split into training/testing with a ratio of 70%/30%. Note that the BIRD benchmark is quite small and contains large-sized formulas with more than 10,000 variables and clauses, it challenges the generalization ability of our model. Besides evaluating in such a data-limited regime, we also conduct experiments on the SATLIB benchmark, an open-source dataset containing a broad range of CNF formulas collected

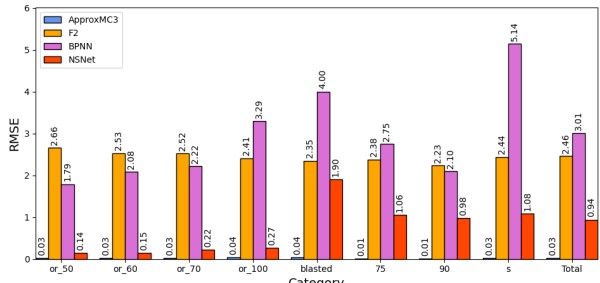 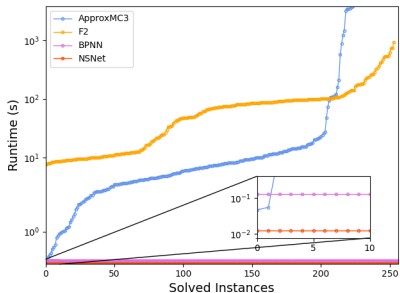

Figure 3: (Left) RMSE between estimated log countings and ground truth for each solver on the BIRD benchmark. (Right) Cactus plots of runtime for each solver on the BIRD benchmark.

from various distributions. To train our model effectively, we choose the distributions with at least 100 satisfiable instances, which include the following 5 categories: (1) uniform random 3-SAT on phase transition region (RND3SAT), (2) backbone-minimal random 3-SAT (BMS), (3) random 3-SAT with controlled backbone size (CBS), (4) "Flat" graph coloring (GCP), and (5) "Morphed" graph coloring (SW-GCP). The whole dataset has 46,200 SAT instances with the number of variables ranging from 100 to 600, and we split it into training/validation/testing sets with a ratio of 60%/20%/20%. For both BIRD and SATLIB benchmarks, we ran the state-of-the-art exact #SAT solver DSharp [30] with a time limit of 5,000 seconds to generate the ground truth labels. The instances where DSharp fails to finish within the time limit are discarded.

**Evaluation & Baselines.** Following BPNN [24], we use the (1) root mean square error (RMSE) between the estimated log countings and ground truth and (2) runtime as our evaluation metrics. We compare NSNet with BP, the neural baseline BPNN [24], and two state-of-the-art approximate model counting solvers, ApproxMC3 [37] and F2 [1]. Akin to SAT solving, both BP and neural models are allowed to run with more iterations until the test metric converges. However, although we try to run BP more iterations to make it coverage, BP consistently fails to achieve comparable results on BIRD and SATLIB benchmarks, so we only leave the other baselines for comparison later. For ApproxMC3 and F2, we set a time limit of 5,000 seconds on each instance. See Appendix B.3.1 for additional setting details and results of these baselines.

**Main Results.** As shown in Figure 3 (Left), NSNet can estimate much tighter countings than BPNN and F2 across all categories of the BIRD benchmark. In total instances, NSNet's estimates are nearly three times preciser than F2's and BPNN's, with a RMSE of 0.94. However, NSNet can not compete with ApproxMC3, whose overall RMSE is only 0.03. We also present the running time of each solver on the BIRD benchmark in Figure 3 (Right). While ApproxMC3 can provide nearly perfect estimates, its time efficiency is undesirable: it spends 123.33 seconds on average for 224 solved ones and fails to finish on the other 33 instances within 5,000 seconds. Unlike ApproxMC3 and F2 running sequentially on CPUs, BPNN and NSNet can run all 257 testing instances parallelly in a batch on a GPU. NSNet is the most time-efficient solver than other baselines, which takes only 4.83 seconds for all problems and 0.02 seconds for each. This demonstrates that NSNet significantly outperforms BPNN and F2 in terms of both the estimate accuracy and time efficiency, and gains more than three orders of speedups over ApproxMC3 while significantly reducing the accuracy gap between them. Additionally, the failure of BP proves that NSNet can leverage the inductive bias of neural networks to synthesize a better message passing algorithm rather than approximating BP.

Table 4 summarizes the performance of each solver on the SATLIB benchmark. We train BPNN multiple times but unsuccessfully make it converge on this dataset, so we discard its performance in the table. Compared with F2, NSNet can provide more precise estimates of the countings in much less time. ApproxMC3 still performs the best among all solvers by a large margin. However, it takes 13.05 seconds on average for each instance, while the number for NSNet is less than 0.01, which is more

Table 4: RMSE and average runtime for each solver on the SATLIB benchmark.

| Method | Metric | |
|---|---|---|
| | RMSE | Runtime (s) |
| ApproxMC3 | **0.05** | 13.05 |
| F2 | 2.36 | 27.79 |
| NSNet | 1.71 | **< 0.01** |

than three orders of magnitude faster than ApproxMC3, showing the efficiency of NSNet. More experimental results on the BIRD and SATLIB datasets can be found in Appendix B.3.2.

# 5   Related Work

**Neural Satisfiability Solvers.**   Using neural networks for satisfiability problems has been explored in the last few years. One common idea is to encode the input formulas using a GNN and perform downstream tasks based on the formulas' embeddings. Recent approaches mainly follow this pipeline and can be classified into two folds [18]: standalone neural solvers and neural-guided solvers.

Standalone neural solvers solve a satisfiability task on their own. NeuroSAT [35] and the following works [22, 10] classify CNF formulas as satisfiable or unsatisfiable. Simultaneously, these methods can also construct a possible assignment by decoding the literal embeddings. Several alternative approaches [2, 3, 31] focus on directly generating satisfying assignments. Such frameworks leverage different GNN architectures and apply an unsupervised loss for training. However, they all attempt to predict a single satisfying solution for each instance, which fails to consider other possible ones.

Although standalone neural solvers show some promising results, neither of these solvers is competitive with the state-of-the-art satisfiability solvers in terms of accuracy or scalability. Hence, another stream of research combines neural networks with modern satisfiability solvers, hoping to improve some components of the existing solvers. These neural-guided solvers include NeuroCore [34] and #Neuro [42], which utilize neural networks to guide the branching decisions of SAT and #SAT solvers. While these works successfully reduce the number of branching steps, frequent invocation of neural networks is also required. By contrast, we call NSNet only once for both SAT and #SAT problems, which significantly diminishes the overhead of querying neural networks. For the SAT problem, NLocalSAT [47] guides the initialization of SLS solvers by the soft assignments of neural networks, which is close to our work. However, they employ the neural architecture similar to NeuroSAT and train it using a specific satisfying assignment, while NSNet is trained to estimate marginals. In addition, there are also some works [45, 25] leveraging reinforcement learning to learn the local search or branching heuristic of SAT solvers.

**Neural Inference Algorithms.**   While BP serves as a non-trainable inference algorithm, some recent works aim to develop learnable inference models, allowing inference to adapt from data. There are some efforts exploring the idea of enhancing BP with neural networks. In particular, [33] proposes NEBP, a hybrid inference method that augments BP by running a GNN co-jointly. BPNN [24] and FE-NBP [40] focus on improving the updates of factor to variable messages by using a learned operator or a learned damping ratio respectively.

Several other methods devise end-to-end inference models beyond the procedures of BP. [46] is the seminal work to apply a GNN to estimate the marginals on relatively small, binary graphical models. FGNN [48] is another GNN model that parameterizes the Max-Product Belief Propagation and performs MAP (maximum a posteriori) inference on factor graphs. Similar to the idea of generalizing BP, FE-GNN [40] leverages the tensor sum operator to the updates of GNNs, which combines factor potentials with message vectors in a similar fashion to BP. Nevertheless, these models are not designed for the case of satisfiability problems and fail to satisfy the negation equivariance of CNF formulas in Proposition 2 while NSNet identifies this property for satisfiability problems.

# 6   Discussion

**Limitations.**   There are a few limitations to NSNet. First, NSNet takes the formulation of solving satisfiability problems as probabilistic inference. However, such a formulation is defined for satisfiable instances but is not applicable to unsatisfiable ones. Therefore, certificating unsatisfiability is beyond the scope of our work. Second, while NSNet can achieve good performance on both SAT and #SAT problems, it requires learning from labeled data, where the labels (especially the ground truth marginals) for hard instances can be time-consuming. Third, although NSNet serves as a neural generalization of BP, it is unclear whether there is a guarantee of the estimated marginals and model counts provided by NSNet. We consider exploring the theoretical guarantee of NSNet as future work.

**Conclusion.** In this work, we present a general neural framework for solving satisfiability problems as probabilistic inference. As a learnable inference model, our framework leverages a novel GNN architecture that subsumes BP in the latent space and performs marginal inference and partition function estimation to solve SAT and #SAT respectively. Experimental evaluations on the synthetic datasets and existing benchmarks demonstrate that our approach significantly outperforms BP and other neural baselines and achieves competitive results compared with the state-of-the-art solvers.

## Acknowledgments and Disclosure of Funding

We thank the anonymous reviewers for their insightful comments. This work was supported, in part, by Individual Discovery Grants from the Natural Sciences and Engineering Research Council of Canada, and the Canada CIFAR AI Chair Program.

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
