# A   Deferred Technical Arguments

## A.1   Explanation and Proof of Proposition 2

We first define the permutation invariance and the negation equivariance of CNF formulas concretely. Given a satisfiable formula $(X_1 \vee \neg X_2) \wedge (\neg X_1 \vee X_3)$, the permutation invariance refers to the fact that all the satisfying assignments are not affected by permuting any variables (e.g. swapping $X_1$ and $X_2$ throughout the formula), by permuting any clauses (e.g. swapping the first clause with the second clause) or by permuting any literals within a clause (e.g. swapping $X_1$ and $\neg X_2$ in the first clause). The negation equivariance means that the assignment of a given variable should be negated if we negate every its corresponding literals (e.g. swap $X_1$ and $\neg X_1$ throughout the formula). Note the permutation invariance and the negation equivariance are important properties of CNF formulas, we also enforce such properties in NSNet.

*Proof.* Given our graph representation of a CNF formula, permuting any variables or clauses is equivalent to changing the orderings of the corresponding assignment nodes or clause nodes in the original graph representation. However, all the learnable modules are not affected by the different orderings, and our summation aggregation and LSE aggregation for updating assignment to clause embeddings and clause to assignment embeddings are also not subject to the orderings of neighboring embeddings. Thus the embeddings between a variable node and an assignment node remain the same after permutation regardless of any parameterization of $h_1$, $h_2$, $\mathcal{A}_1$, $\mathcal{A}_2$ and $\mathcal{A}_3$. On the other hand, permuting any literals within a clause has no effect on our graph construction, so all of the embeddings in NSNet remain unchanged. Similarly, negating every literal corresponding to a given variable $X_i$ is equivalent to swapping two assignment nodes $X_i^0$ and $X_i^1$ in our graph representation, while all the edges in the graph remain the same. Therefore, all the edge embeddings connected to $X_i^0$ are the same as the embeddings connected to $X_i^1$ after negating and vice versa.   $\square$

## A.2   Constructing A Satisfying Assignment from Marginals

Besides performing the stochastic local search, there are multiple ways to generate a satisfying assignment from the estimated marginals. One common approach is the decimation algorithm [29] (Algorithm 1), which processes the following steps iteratively: (1) estimate the variable marginals. (2) fix a variable with the highest certainty (whose marginal value is the most extreme) to the value 0 or 1. (3) simplify the given formula. If we can estimate the marginals accurately at each iteration, such a process would act as an oracle search without backtracking to construct a satisfying assignment. Besides the decimation algorithm, one can also combine NSNet with backtracking search solvers by using the estimated marginals to guide the branching heuristic in these solvers. However, if we integrate NSNet with the decimation algorithm or the backtracking-based solvers, each iteration of these processes needs to query the neural networks on a new simplified formula, which is computationally demanding and impractical for large instances. To reduce the overhead of querying neural networks, we call NSNet only once to estimate marginals and execute a local search to find a satisfying assignment.

---

**Algorithm 1** The decimation algorithm

---

**Input:** A satisfiable formula $\Phi$ with $n$ variables
  1: $\Phi_0 \leftarrow \Phi$
  2: **for** $t \leftarrow 1$ to $n$ **do**
  3:      Estimate marginals $b_i(1), b_i(0)$ for each variable $X_i$ of the formula $\Phi_{t-1}$
  4:      Find the variable $X_j$ with the highest value $|b_j(1) - b_j(0)|$
  5:      **if** $b_j(1) > b_j(0)$ **then**
  6:          $x_j \leftarrow 1$
  7:      **else**
  8:          $x_j \leftarrow 0$
  9:      **end if**
10:      Obtain a new formula $\Phi_t$ from $\Phi_{t-1}$ by substituting $x_j$ for variable $X_j$ and simplifying
11: **end for**
12: **return** The assignment $\boldsymbol{x} = \{x_1, x_2, \ldots, x_n\}$

---

# B  Additional Experimental Details

## B.1  Implementation Details

For training, we use the Adam optimizer [23] with a learning rate of 1e-4 and a weight decay of 1e-10 and clip the gradient with a global norm of 0.65. We train all the neural networks with a batch size of 128 for 200 epochs on synthetic datasets and 1000 epochs on the BIRD and SATLIB benchmarks. For experiments on the synthetic datasets and SATLIB benchmark, we select the best checkpoint for each model based on its performance on the validation set. We run BPNN using the official code [2], and implement NeuroSAT and NSNet using PyTorch [32] and PyTorch Geometric [16].

## B.2  SAT Solving

### B.2.1  Datasets

For SR, we use the same parameters as NeuroSAT but limit the maximum length of each clause to 4. For random 3-SAT, we generate satisfiable instances where the relationship between the number of clauses ($m$) and variables ($n$) is $m = 4.258n + 58.26n^{-\frac{2}{3}}$ [13]. For CA, we set the number of communities between 3 to 10 and the modularity factor $Q$ between 0.7 and 0.9. Note that the $Q$ value is typically less than 0.3 for random k-SAT problems but larger than 0.7 for real-world instances [4].

### B.2.2  Results

We also test the performance of the SLS solvers by reporting their average number of flips. As shown in Table 5, all modified Sparrow solvers can use fewer flips than Sparrow to find a satisfying solution while solving much more instances at the same time. This further demonstrates the effectiveness of the initial assignments from the estimated marginals. Among these SLS solvers, NSNet-Sparrow can not only solve more instances than other SLS solvers but also generate a satisfying assignment with the least local search steps.

Table 5: Average number of flips for Sparrow with different initializations on the synthetic datasets. We only take the solved instances into account.

| Method | Larger Distribution | | | |
|---|---|---|---|---|
| | SR | 3-SAT | CA | Total |
| Sparrow | $60.68 \pm 0.80$ | $58.59 \pm 0.46$ | $56.82 \pm 0.26$ | $57.55 \pm 0.21$ |
| BP-Sparrow | $19.32 \pm 0.43$ | $17.27 \pm 0.25$ | $18.76 \pm 0.15$ | $18.51 \pm 0.15$ |
| NeuroSAT-Sparrow | $26.45 \pm 0.55$ | $22.59 \pm 0.51$ | $18.87 \pm 0.20$ | $20.91 \pm 0.22$ |
| NSNet-Sparrow | $\mathbf{16.28 \pm 0.25}$ | $\mathbf{14.16 \pm 0.27}$ | $\mathbf{15.86 \pm 0.18}$ | $\mathbf{15.53 \pm 0.11}$ |

## B.3  Model Counting

### B.3.1  Baselines

To ensure a fair comparison, we train BPNN using the message passing iteration of 10 rather than 5 in its original paper, which also slightly improves its performance on the BIRD benchmark. Note that ApproxMC3 provides probably approximately correct (PAC) guarantee on the estimated model count with two parameters: the tolerance $\epsilon$ and the confidence 1-$\delta$, we first run ApproxMC3 in the default settings ($\epsilon$=0.8, $\delta$=0.2) and further conduct experiments with different settings. Following the evaluation of BPNN, we run F2 with the default parameters and choose the low bound mode. In addition, for these two solvers, finding *minimal independent support* (MIS) [20] is always used as a preprocessing step to boost their computations. So we also use the MIS tool [21] within 1k seconds as the preprocessing. We record two times for each instance: one is the sum of the MIS tool's runtime and the time of the #SAT solvers with the MIS support; the other is the running time of the #SAT solvers without the MIS. The minimum of these two times is reported. For BP, we try to perform message passing with $T = 10, 20, 50, 100, 200, 500$ iterations, but only achieve the best overall RMSE on the BIRD benchmark and SATLIB benchmark of 20.42 and 17.67 respectively, which is not comparable with other baselines.

---

[2] https://github.com/jkuck/BPNN.

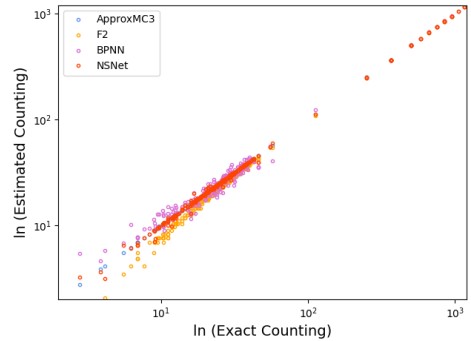 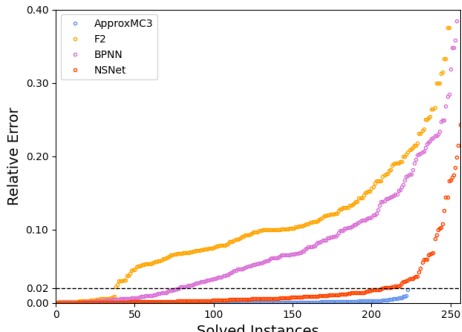

Figure 4: (Left) Scatter plot comparing the estimated log countings against the ground truth for each solver on the BIRD benchmark. (Right) Relative error between the estimated log countings and the ground truth log countings for each solver on the BIRD benchmark.

### B.3.2 Results

Figure 4 (Left) shows the scatter plot comparing the estimated log countings against the ground truth for each solver on the BIRD benchmark. We can observe that both ApproxMC3 and NSNet can provide tighter estimates than both F2 and BPNN on most instances when the ground truth is less than $e^{100}$. While ApproxMC3 fails to finish in 5,000 seconds when the ground truth counting is more than $e^{100}$, NSNet can still give tight approximations when the ground truth counting is even more than $e^{1,000}$. This demonstrates the effectiveness of NSNet to solve hard and large instances. We further report the relative error between the estimated log countings and the ground truth log countings in Figure 4 (Right). On average, NSNet's relative error is less than 2%, which is significantly better than F2's and BPNN's. Note that NSNet only spends 0.02 seconds for each instance, such relative error is also acceptable in many applications.

Table 6 shows the detailed RMSE results of each solver on the SATLIB benchmark. Compared with its performance on the BIRD benchmark, the precision of NSNet decreases by a large margin. We conjecture this is because the data of the BIRD benchmark is collected from many real-world model counting applications, which may share a lot of common logical structures to learn. On the other hand, the instance in the SATLIB benchmark is generated randomly, making NSNet hard to exploit common features. Nevertheless, NSNet still outperforms F2 in most categories.

Table 6: RMSE between estimated log countings and ground truth for each solver on the BIRD benchmark.

| Method | Distribution | | | | | |
|--------|---------|------|------|------|--------|-------|
|        | RND3SAT | BMS  | CBS  | GCP  | SW-GCP | Total |
| ApproxMC3 | 0.04 | 0.05 | 0.05 | 0.06 | 0.05 | 0.05 |
| F2     | 2.13    | 2.42 | 2.37 | 2.40 | 2.66   | 2.36  |
| NSNet  | 1.57    | 2.45 | 1.68 | 2.14 | 1.37   | 1.71  |

Since ApproxMC3 can be configured to achieve different trade-offs between speed and accuracy, we also test it with different settings. Table 7 shows the performance of ApproxMC3 with different $\epsilon$ and $\delta$. Although we significantly relax the theoretical PAC guarantee on the estimated model count to improve the speed of ApproxMC3, ApproxMC3 can still give quite tight estimates while spending orders of magnitudes time than NSNet in practice. Additionally, ApproxMC3 timeouts on more than 30 instances in 5,000 seconds while NSNet solves all the instances. We believe the overhead of ApproxMC3 is still significant with much loose bound because it needs to frequently call the CryptoMiniSat [38] to reason about subformulas of the original CNF formula. Instead, NSNet only performs message passing to provide an estimation, which is much more efficient. To trade the slight inaccuracy with significant speedup, NSNet can serve as a more feasible choice.

Table 7: RMSE between estimated log countings and ground truth for ApproxMC3 with different parameters on the BIRD benchmark.

| Parameter | | Metric | | |
|---|---|---|---|---|
| $\epsilon$ | $\delta$ | RMSE | Avg. runtime (s) | #Failed |
| 0.8 | 0.2 | 0.03 | 123.32 | 33 |
| 0.8 | 0.8 | 0.13 | 39.07 | 33 |
| 4 | 0.2 | 0.07 | 63.95 | 33 |
| 4 | 0.8 | 0.21 | 23.69 | 34 |
| 10 | 0.99 | 0.23 | 22.45 | 35 |