# OpenReview forum: "NSNet: A General Neural Probabilistic Framework for Satisfiability Problems"
_NeurIPS.cc/2022/Conference — NeurIPS 2022 Accept_

### Official Review · Reviewer_n673 · 2022-07-08

**Rating:** 6
**Confidence:** 3
**Soundness:** 2 fair
**Presentation:** 3 good
**Contribution:** 2 fair

**Summary:**

The paper proposes NSNet, a novel graph neural network (GNN) framework taking the SAT solving problems as probabilistic inference problems. It proposes a new graph encoding for SAT formulas and employs a novel message passing mechanism to parameterize Belief Propagation in the latent space. Because of BP, NSNet can estimate both the marginals and the partition function of graphical models, which make it applicable to solve both SAT and #SAT problems. In particular, for solving SAT problems, NSNet can be trained in an unsupervised manner; for solving #SAT problems, NSNet is trained in a supervised manner.

**Questions:**

1. What is the reason to make NSNet-Sparrow less time-efficient than Sparrow?
2. Can NSNet be scalable to large SAT problems with thousands and millions of variables, such as the ones in SAT competitions?
3. How effective is NSNet in solving unsatisfiable problems?
4. Is there any theoretical guarantee for the accuracy of NSNet in approximating the model count?

**Limitations:**

The authors point out two limitations of NSNet. One is that NSNet cannot be applicable to unsatisfiable problems. However, this limitation makes NSNet not practical for solving real-world problems. Because one of the important functionality of SAT solvers is to check the satisfiability for both SAT and UNSAT problems. The second limitation is that NSNet has to be trained with supervised learning. The authors point potential research direction for solving this limitation which is inspiring for future research.

**Strengths And Weaknesses:**

Strengths:
1. The paper is trying to solve two important problems (i.e., SAT and #SAT) which can impact many fields in computer science such as security, verification, etc.
2. The method of utilizing a novel GNN that parametrizes BP in the latent space to do marginal inference among all satisfying solutions is interesting.
3. The experimental results of NSNet in approximating model counting are promising.

Weaknesses:
1. The experimental results of NSNet in solving SAT problems are not good enough for practical usage. In addition, NSNet is not applicable for unsatisfiable problems.
2. No theoretical guarantee provided for the accuracy of NSNet in approximating the model counts.

Overall the motivation of the paper is convincing. It is trying to solve two important problems that can impact various kinds of fields in computer science. The idea of utilizing GNN to do marginal inference among all satisfying solutions for solving both SAT and #SAT problems is novel.

For solving SAT problems, NSNet can be trained in an unsupervised learning fashion without requiring labels of the data samples. However, I cannot be convinced by the experimental results that NSNet can be a preferable choice comparing to classic SAT solvers. First, the benchmarks utilized in the experiment are only form SATLIB. However, the instances in SATLIB are the ones that are small and relatively easy to solve. As mentioned in the paper, the number of variables in these benchmarks range from 100 to 600. However, the benchmarks in the SAT competition and the ones produced by practical applications can easily have thousands or even millions of variables. In addition, the runtime results in the appendix show that the SLS solver Sparrow takes only 245.44 seconds for solving all 9,260 testing instances, with an average of 0.02 seconds to solve each instance. From the results, I can tell that these benchmarks are easy to solve. I would recommend to do the experiments on the SAT competition benchmarks which are the one experimented by the NeuroCore solver (Selsam, Daniel, and Nikolaj Bjørner. "Guiding high-performance SAT solvers with unsat-core predictions." International Conference on Theory and Applications of Satisfiability Testing. Springer, Cham, 2019).

Besides, the authors only show that NSNet can outperform the state-of-the-art solver Sparrow in the solving accuracy. In the real world, what users of SAT solvers really care about is the time efficiency. However, as shown in the appendix, the solving runtime of NSNet-Sparrow is less efficient than the SLS solver Sparrow. This makes the current state of NSNet not practical for real-world usage.

For approximate model counting, the experimental results show that NSNet is able to outperform the state-of-the-art approximate model counter ApprxoMC in time efficiency. This makes NSNet a promising approach to consider in approximating the model count for SAT problems. However, the drawback of NSNet is that the approximation accuracy is not as good as ApproxMC. Besides, only the average of the counting accuracy is shown in the experiment. It could provide the readers with more insights if the authors can also provide the minimum and maximum approximation accuracy. In addition, I wonder if the authors can provide any theoretical guarantees/proofs about the approximate accuracy.

---

> ### Author Response · Authors · 2022-08-02
> **Response to Reviewer n673**
>
>
> Thank you for the positive feedback and in-depth comments. In the following, please let us address your raised concerns and questions.
>
> ### Response to the questions
>
> > What is the reason to make NSNet-Sparrow less time-efficient than Sparrow?
>
> NSNet-Sparrow is less time-efficient than Sparrow because NSNet-Sparrow takes some extra time of querying NSNet into accounts. In our evaluation on the SATLIB benchmark, the average of querying NSNet is roughly 0.01 seconds per instance, which we believe is a modest overhead. In the case a SAT instance is challenging, the time of the local search procedure becomes dominant and the overhead of querying NSNet would be negligible. On the other hand, better engineering efforts and stronger hardware would make NSNet more efficient.
>
> > Can NSNet be scalable to large SAT problems with thousands and millions of variables, such as the ones in SAT competitions?
>
> For the SAT datasets, existing neural SAT solvers mainly conduct experiments on small synthetic instances (#vars < 100) with a specific distribution. As stated in our paper, we choose the SATLIB benchmark because it has much more challenging and diverse instances than existing synthetic datasets to make NSNet more practical. We agree that there is still a scalability gap between these instances and the instances in SAT competitions. Indeed, we also considered using the instances in SAT competitions, but there are several limitations of using SAT competitions to train NSNet:
>
> - The number of instances in SAT competition is not sufficient for unsupervised training of NSNet (hundreds of instances every year and some of them are republicated). To make NSNet capable of generalizing unseen instances well, one needs to train NSNet on sufficiently large datasets.
>
> - Some instances of SAT competition have millions of variables and clauses that cannot fit in our single GPU memory. As stated in the paper [1], NeuroCore is trained on a cluster with 20 GPUs. Due to the limited computing resources, we did not conduct experiments on SAT competitions.
>
> In addition, we also plan to request more computing resources and collect large datasets in domain-specific applications (e.g. bounded model checking) to test and improve NSNet in the future.
>
> [1] Selsam, Daniel, and Nikolaj Bjørner. "Guiding high-performance SAT solvers with unsat-core predictions." _International Conference on Theory and Applications of Satisfiability Testing_. Springer, Cham, 2019.
>
> > How effective is NSNet in solving unsatisfiable problems?
>
> The probabilistic formulation of NSNet is constructed on the assumption that the given CNF formula is satisfiable. We conjecture that all GNN architectures can only “solve” the satisfiable instances rather than unsatisfiable ones (predicting unsatisfiability without proof is relatively trivial, which we don’t consider as “solving”). For satisfiable instances, existing GNN frameworks may help to find a satisfying assignment. But for unsatisfiable ones, the neural networks need to prove that all of the possible assignments cannot satisfy the formula or construct an unsat proof to certify the unsatisfiability. However, such a procedure needs to back-tracking search, making the existing GNN frameworks hard to learn by performing massage passing. Indeed, the paper [2] gives a more systematic justification. In other words, all the (standalone) neural SAT solvers are incomplete solvers that are hard to "solve" unsatisfiable instances. To make NSNet able to "solve" unsatisfiable instances (with proofs), one can combine NSNet with existing complete solvers.
>
> [2] Chen, Ziliang, and Zhanfu Yang. "Graph neural reasoning may fail in certifying boolean unsatisfiability." _arXiv preprint arXiv:1909.11588_ (2019).
>
> > Is there any theoretical guarantee for the accuracy of NSNet in approximating the model count?
>
> For BP, it is guaranteed to return the exact model count if the underlying graphical model of an instance is a tree. When applied to models with cycles, the fixed point of BP is only guaranteed to be a local minimum of the Bethe free energy. By subsuming BP in the latent space, NSNet severs as a learnable inference algorithm that is likely to inherit the theoretical guarantee of BP. Although it is not yet clear whether NSNet can provide the PAC-style guarantee like ApproxMC3, we show the empirical evidence and strong performance of NSNet in our evaluation on the BIRD benchmark: after training, it can provide quite tight estimates (relative error <2%) while being orders of magnitude faster than ApproxMC3.

---

> > ### Author Response · Authors · 2022-08-02
> > **Response to Reviewer n673 (cont.)**
> >
> > ### Clarification of the weaknesses
> >
> > > The experimental results of NSNet in solving SAT problems are not good enough for practical usage.
> >
> > We agree that there is still a significant gap between purely neural SAT solvers and the modern CDCL-based SAT solvers. To close this gap, progresses in understanding the capability and the limitations of neural-based approaches are well-needed. NSNet is particularly interesting because it shows that neural network is well-suited (perhaps better suited) for learning variable marginals. Simply rounding the estimated marginals by NSNet (without local search) is already more effective than directly predicting a satisfying solution. As discussed in the appendix, with the estimated marginals, it is also flexible to combine NSNet with CDCL solvers or the decimation algorithm to obtain a satisfying assignment. However, to make such a combination efficient for practical usage, one also needs more engineering efforts and computing resources to improve the time efficiency of querying the neural networks. We believe the hope of outperforming modern state-of-the-art SAT solvers is to combine learnable components like NSNet with carefully designed modern solvers.

---

> > ### Comment · Reviewer_n673 · 2022-08-09
> > **Keeping my original scores**
> >
> > Thank to the authors for answering my questions. The scalability of NSNet and the lack of approximation gaurantees of NSNet are the two main limitations of this paper, for which I hope the authors would try to improve/discuss in the final paper version.

---

### Official Review · Reviewer_bWLw · 2022-07-10

**Rating:** 7
**Confidence:** 4
**Soundness:** 3 good
**Presentation:** 3 good
**Contribution:** 3 good

**Summary:**

This paper introduces NSNet, a GNN-style neural network approach to improve belief propagation (BP). GNN networks have previously been used for SAT, #SAT and QBF, but here it is paired with insights from the BP algorithm. The experimental evaluation demonstrates clear gains over the existing BPNN and F2 methods on #SAT in terms of prediction error and runtime. It also improves over NeuroSAT on plain SAT in terms of the number of solved instances. The paper also claims gains over ApproxMC3 which are not supported by the evidence.

**Questions:**

Please respond to/refute the concerns raise in "strengths and weaknesses".

**Limitations:**

No immediate societal impact.

**Strengths And Weaknesses:**

Impact: This paper presents significant progress in cheap estimates of #SAT instances with belief propagation supported by GNNs, which is a strong result! The progress on plain SAT is probably irrelevant in practical terms, as there are much better exact SAT solvers. I also highly appreciate the clear statement on limitations of the approach.

Unfortunately the paper includes some claims with respect to ApproxMC3 that I do not find convincing. I kindly ask the authors to carefully inspect these issues and either explain how I am wrong or revise the claims. The score I gave the paper reflect my current perception of the paper and I'd be happy to be proven wrong and will increase my score accordingly.

Questionable claim: "This demonstrates that NSNet [...] gains more than three orders of speedups over ApproxMC3 at an acceptable cost of precision." This paper does not contain any evaluation about the impact of the precision loss on downstream applications, so calling them "acceptable" is not appropriate. Instead the authors should emphasize that they significantly reduce the accuracy gap between ApproxMC3 an the competing methods.

Benchmarks: There are established #SAT benchmarks that are used for the evaluation of ApproxMC3. This paper does not even mention them, which is a serious omission, given the claims of gains over ApproxMC. Instead, the benchmarks chosen here are purely synthetic formulas. These synthetic formulas have no direct relation to #SAT as far as I can see, so I am even somewhat cautious about the comparison between BPNN, F2, and NSNet.

Comparison to ApproxMC: As far as I know, the ApproxMC solvers can be configured to achieve different trade-offs between speed and accuracy. The paper here appears to not consider that and instead claims gains in speed while trailing in accuracy. In other words, ApproxMC might be much faster if it is not forced to produce very exact results. I acknowledge that it might be hard to find an apples-to-apples comparison. But I believe that a better evaluation is necessary to support the claims made in the paper.

=========
Update after the rebuttal and discussion phase: The authors were able to address my concerns and I have to revise my view of the paper. The results on #SAT are pretty impressive. This is a method to quickly estimate the solution count of complicated formulas. Unfortunately the approach does not provide any guarantees on the accuracy, but this may still be very helpful in the field.

I do not share the concerns about novelty raised by some other reviewers. While the method differs only in details from other published approaches, these details appear to be pretty important as the experiments suggest.

---

> ### Author Response · Authors · 2022-08-02
> **Response to Reviewer bWLw**
>
>
> Thank you for the detailed comments and questions. In the following, we hope to address the stated weaknesses of our paper.
>
> > The progress on plain SAT is probably irrelevant in practical terms, as there are much better exact SAT solvers.
>
> We agree that there is still a significant gap between purely neural SAT solvers and the modern CDCL-based SAT solvers. To close this gap, progresses in understanding the capability and the limitations of neural-based approaches are well-needed. NSNet is particularly interesting because it shows that neural network is well-suited (perhaps better suited) for learning marginals. Simply rounding the estimated marginals by NSNet (without local search) is already more effective than directly predicting a satisfying solution (as shown in Table 1). We believe the hope of outperforming modern state-of-the-art SAT solvers is to combine learnable components like NSNet with carefully designed solvers and algorithms, such as CDCL solvers, SLS solvers, or the decimation algorithm. We have discussed several promising application scenarios for using the estimated marginals of NSNet in the appendix.
>
> > This paper does not contain any evaluation about the impact of the precision loss on downstream applications, so calling them "acceptable" is not appropriate.
>
> We agree that the wording "acceptable" is not very accurate, since as the reviewer pointed out, whether the approximation accuracy is acceptable would be application specific. By using “acceptable” in our writing, we mean 1) the accuracy of NSNet is fairly close (i.e., relative error <2%) to the ground truth and is significantly better than BPNN and F2. 2) NSNet can still provide quite tight estimations in less than 1 second on many hard instances where ApproxMC3 fails to give an output in 5,000 seconds. To make our claims more precise and accurate, we will clarify this in the revision of this paper.
>
> > #SAT Benchmarks
>
> For #SAT, we evaluate NSNet and peer methods on **two** datasets: **BIRD** benchmark and SATLIB benchmark. As discussed in our experimental setup in Section 4.2, the BIRD benchmark is an established #SAT benchmark that was originally provided in the paper of ApproxMC3 [1]. We use “BIRD” to refer to this benchmark because of the paper title of ApproxMC3. The used BIRD dataset consists of 862 real-world instances collected from different applications. Some of them have more than 10,000 variables and clauses with more than e^10,000 solutions, which is still challenging for ApproxMC3. As stated in our paper, ApproxMC3 fails on 33 instances in 5,000 seconds on the testing set.
>
> [1] Soos, Mate, and Kuldeep S. Meel. "BIRD: engineering an efficient CNF-XOR SAT solver and its applications to approximate model counting." _Proceedings of the AAAI Conference on Artificial Intelligence_. Vol. 33. No. 01. 2019.
>
> > More configurations of ApproxMC3
>
> ApproxMC3 provides probably approximately correct (PAC) guarantee on the estimated model count with two parameters: the tolerance $\epsilon$ and the confidence 1-$\delta$. In our paper, we run ApproxMC3 in the default settings ($\epsilon$=0.8, $\delta$=0.2). We further run ApproxMC3 with different $\epsilon$ and $\delta$ on the BIRD benchmark with a time limit of 5,000 seconds. The results are shown in the below table:
>
> | $\epsilon$ | $\delta$ | RMSE | avg. time (s) | #failed |
> |------------|----------|------|-------------|---------|
> | 0.4        | 0.1      | 0.01 | 56.57       | 40      |
> | 0.8        | 0.2      | 0.03 | 123.32      | 33      |
> | 0.8        | 0.8      | 0.13 | 39.07       | 33      |
> | 4          | 0.2      | 0.07 | 63.95       | 33      |
> | 4          | 0.8      | 0.21 | 23.69       | 34      |
>
> When we relax the theoretical PAC-guarantee on the estimated model count to improve the speed of ApproxMC3, ApproxMC3 can still give quite tight estimates while spending orders of magnitudes time than NSNet in practice. Although the average running time of ApproxMC3 on the solved instances decreases as the theoretical bounds relax, ApproxMC3 still fails to return in 5,000 seconds on many instances. This is because ApproxMC3 needs to frequently call the CryptoMiniSat solver to reason about subformulas of the original CNF formula while NSNet only needs to perform message passing to provide an estimation, which is much more efficient.

---

> > ### Comment · Reviewer_bWLw · 2022-08-05
> > **Clarifications**
> >
> > Thank you for the additional experiments and explanations.
> >
> > BIRD benchmark: Sorry, my fault. I did not notice that is the benchmark from the ApproxMC3 paper.
> >
> > Additional questions:
> > - In the rebuttal, you mentioned that the "relative error" is <2%, though the RMSE is ~2. I read this as a factor of 2, not as 2%. Could you elaborate?
> > - In your additional experiments where you lowered the precision of ApproxMC3, the precision dropped by a factor of 7 (RMSE 0.03 -> 0.21) and the runtime dropped by a factor of ~5. However, the RMSE here is still a factor of 10 better than what NSNet can produce. If this trend of RMSE vs avg time continues to scale in a similar factor, ApproxMC3 might be similarly fast as NSNet on many formulas at a comparable RMSE. I am not convinced these experiments fully support the point that you made in the rebuttal.
> > - Did you also compare to ApproxMC4, which seems to offer better performance than ApproxMC3? (I think this is not strictly necessary, but it would make the paper stronger.)

---

> > > ### Author Response · Authors · 2022-08-07
> > > **Further responses**
> > >
> > > Thank you for the quick follow-up.
> > >
> > > > RMSE and relative error
> > >
> > > Following the evaluation of BPNN, we use RMSE as our metric to evaluate the accuracy of the approximate #SAT solvers. On average, the RMSE of NSNet on the BIRD dataset is 0.94 (Figure 3 Left). Note that RMSE describes the "absolute error" between the estimated _log_ model count and the ground truth, we further evaluate the "relative error" between them. As shown in the appendix, the relative error of NSNet on the BIRD dataset is less than 2%.
> > >
> > > > Performance of ApproxMC3
> > >
> > > We relax the theoretical guarantee of ApproxMC3 to improve its speed but find that it still provides quite precise estimates while spending a lot of time. To further demonstrate our claims, we run ApproxMC3 with more parameters. The results are shown in the below table:
> > >
> > > | $\epsilon$ | $\delta$ | RMSE | avg. time (s) | #failed |
> > > |------------|----------|------|-------------|---------|
> > > | 10 | 0.99 | 0.23 | 22.45 | 35 |
> > > | 100 | 0.99 | 0.24 | 22.46 | 35 |
> > > | 1,000 | 0.99 | 0.24 | 21.36 | 35 |
> > > | 10,000 | 0.99 | 0.24 | 23.57 | 35 |
> > >
> > > Although we significantly relax the theoretical guarantee of ApproxMC3 by varying $\epsilon$ and $\delta$, we don't observe the hypothesized trend (as the reviewer suggested). Additionally, ApproxMC3 still timeouts on 35 instances while NSNet solves all the instances. We believe the overhead of ApproxMC3 is still significant with much loose bound because it ultimately relies on the CryptoMiniSat solver. To trade the slight inaccuracy with significant speedup, NSNet can serve as a more feasible choice.
> > >
> > > > Performance of ApproxMC4
> > >
> > > Following the evaluation of BPNN, we choose ApproxMC3 as our baseline. We further run ApproxMC4 on the BIRD dataset with different parameters. The results are shown in the below table:
> > >
> > > | $\epsilon$ | $\delta$ | RMSE | avg. time (s) | #failed |
> > > |------------|----------|------|-------------|---------|
> > > | 0.4 | 0.1 | 0.03 | 237.13 | 15 |
> > > | 0.8 | 0.2 | 0.03 | 234.77 | 15 |
> > > | 0.8 | 0.8 | 0.03 | 237.99 | 14 |
> > > | 4 | 0.2 | 0.03 | 235.15 | 14 |
> > > | 4 | 0.8 | 0.03 | 257.43 | 11 |
> > > | 10,000 | 0.99 | 0.03 | 248.93 | 11 |
> > >
> > > ApproxMC4 can provide much tighter approximations than ApproxMC3 with RMSE of 0.03 consistently, although we relax its theoretical guarantee. It can also solve more instances in 5,000 seconds than ApproxMC3, but it still fails to return on more than 10 instances of all 257 testing instances. On the instances solved by both ApproxMC3 and ApproxMC4, ApproxMC4 spends 20.60 seconds on average (when $\epsilon=10,000$ and $\delta=0.99$), which is indeed faster than ApproxMC3.
> > >
> > > We will include this in the future revision of our paper and are happy to hear further questions or feedback.

---

> > > > ### Comment · Reviewer_bWLw · 2022-08-08
> > > > **Thanks for the additional clarifications**
> > > >
> > > > Thank you for addressing my concerns. I'm revising my view of the paper - this is good work! Will change my score for the paper accordingly.

---

### Official Review · Reviewer_8cDU · 2022-07-11

**Rating:** 3
**Confidence:** 4
**Soundness:** 3 good
**Presentation:** 3 good
**Contribution:** 2 fair

**Summary:**

The paper proposes a neural model for the tasks of SAT and #SAT. For SAT solving, the paper proposes to learn marginal distributions and combine them with local search to get a feasible assignment. For #SAT, the paper learns the Bethe approximation of the partition function. Experiments verify the effectiveness of the proposed model.

**Questions:**


(1) Please give in-depth discussions on how your work is different from existing works regarding model architecture and learning losses.
(2) Regarding learning losses, what new perspectives do we have? Simply applying existing/similar losses to learn a model is not good enough for NeurIPS. There is a similar work [1] that also uses the Bethe approximation of the partition function for #SAT.

[1] https://arxiv.org/abs/2205.04423

**Limitations:**

N/A.

**Strengths And Weaknesses:**

Strengths:

The paper proposes to learn NN models for solving SAT and #SAT with unsupervised learning.

Weaknesses:

The contributions are incremental. Combing GNN with BP is not new, and the losses to learning the model are not novel.

---

> ### Author Response · Authors · 2022-08-02
> **Response to Reviewer 8cDU**
>
>
> Thank you for reviewing our paper. Nevertheless, it seems there are some misunderstandings between NSNet and existing works. In the following, please let us clarify the difference between NSNet and other methods more clearly.
>
> > Comparison between NSNet and existing neural solvers
>
> - Compared with existing neural SAT solvers, NSNet is the first one that takes the formulation of probabilistic inference and performs marginal inference rather than directly predicting a satisfying assignment. Its GNN architecture leverages a novel graph representation and message passing scheme that explicitly incorporates BP in the latent space, whereas other neural SAT solvers use common GNN architectures (e.g. gated graph neural network). NSNet can also be flexibly configured to perform partition function estimation for #SAT while existing neural SAT solvers can not be used in that way since they cannot compute the factor beliefs as BP.
>
> - Compared with the existing neural version of BP, NSNet is designed for both SAT and #SAT problems by performing marginal inference and partition function estimation respectively, while prior works mainly focus on only a single inference task (e.g. MAP inference). NSNet generalizes BP in the latent space (i.e. each message is a high dimensional vector rather than a scalar) while other methods (e.g. BPNN) only modify some scalars (e.g. damping ratio) of BP and maintain the main procedure of BP. Furthermore, NSNet satisfies the negation equivariance of CNF formulas while other neural versions of BP fail to consider this property. These make NSNet more powerful to learn a better inference algorithm beyond BP.
>
> > Novelty and new perspective of the learning losses
>
> - For SAT, NSNet uses an interpretable unsupervised loss from a Bayesian perspective, whereas prior works use different losses and don’t explore such a probabilistic formulation. With this loss, NSNet is trained to perform marginal inference while existing neural SAT solvers are trained to predict a specific satisfying assignment directly. Furthermore, to compare these two learning objectives, we also train the neural SAT solvers with the supervision of a specific satisfying assignment to directly predict an assignment. Experiments on the SATLIB benchmark show that the supervision in this way is problematic and leads to quite low solving accuracy. Instead, using unsupervised learning to predict the variable marginals is more suitable for neural SAT solvers and achieves higher solving accuracy.
>
> - For #SAT, NSNet uses the ground-truth label since the solution to a #SAT problem is a deterministic value. However, as discussed in the limitation of our paper, we believe it is possible to train the neural network without supervision since BP doesn't need such supervision to estimate the model count. We leave it as a future direction.
>
> > A similar work on arXiv
>
> Thanks for pointing out the recent related work BPGAT on #SAT solving. Since it was only uploaded to arXiv near the NeurIPS paper deadline (and not published anywhere to our best knowledge), we unfortunately did not notice this work. After a careful reading, we notice some important differences between NSNet and BPGAT. Similar to BPNN, BPGAT maintains the procedure of BP but only uses a very simple neural network ("through a single-layer feedforward neural network **a**  $\in \mathbb{R}^4$", see page 6 of BPGAT) to learn a weighted summation aggregation in BP. In contrast, NSNet is a GNN architecture that generalizes the entire BP procedure in the latent space and introduces more inductive bias for both SAT and #SAT problems, which can learn a much more powerful inference algorithm beyond BP. We will cite this recent work in the future revision of our paper.

---

### Official Review · Reviewer_Tesa · 2022-07-13

**Rating:** 5
**Confidence:** 5
**Soundness:** 3 good
**Presentation:** 3 good
**Contribution:** 2 fair

**Summary:**

This paper provides a neural framework for SAT solving based on Belief Propagation. The authors propose a GNN architecture that captures BP in the latent space and they perform marginal inference using the trained model to obtain satisfying assignments for SAT. They also use the partition function estimation to approximately solve the model counting version of SAT, #SAT. The task for SAT is to find a satisfying assignment to problems that are known to be satisfiable. Equipped with this fact they manage to train the model in an unsupervised way by maximizing the log-likelihood of the model's prediction of being 1 (i.e., SAT). For #SAT they minimize the RMSE of model’s prediction and the true model count of a given formula.

They experimentally show that their approach outperforms vanilla BP and other neural approaches on SATLIB and BIRD benchmarks. For SAT they further experiment with initializing an SLS solver to obtain a true satisfying assignment.

**Questions:**

Given the close similarities between the proposed approach and some of the baselines used in the paper, it is interesting to have an intuitive explanation as to why NSNet achieves better results relative to other neural solvers, such as NeuroSAT and BPNN? Specifically a more thorough ablation and architectural comparison to BPNN is very welcome! Some example metrics to compare the models are number of parameters, number of inference time message passing iterations used for NeuroSAT and BPNN, etc.

**Limitations:**

See above

**Strengths And Weaknesses:**

Strengths:
========
The paper is well-written and well-motivated and has a principled way of deriving the GNN architecture from BP. The

Weaknesses:
===========

[Novelty:]
The main issue with the paper is with the novelty of the proposed approach:
- The use of BP in the context of SAT has been extensively explored in the past but in a neural and non-neural fashion.
- The Neural SAT solvers although not directly designing their model architecture from BP, implicitly perform similar type of computation as NSNet.
- Training the model for SAT in an unsupervised way using equation (9) is somewhat novel although the label of the instance still needs to be known a priori since the model is trained on only satisfiable instances.
- Obtaining a satisfying assignment from variable biases is not novel as NeuroSAT also extracts satisfying assignments by training on a single bit of supervision.
- Lastly initializing an SLS solver with neural predictions is also not novel and has been done by Li et al. [1] and others in the past.

On the #SAT front the use of variable biases and factor beliefs to predict the log of the model count of the given formula was also done by Kuck et al. in BPNN.

[Results:]
The results are interesting and the proposed approach does indeed better on the chosen baselines in terms of solving accuracy of the initial assignments for SAT. When it comes to initializing the SLS solvers however, the results are more or less on par with vanilla BP (BP-Sparrow vs. NSNet-Sparrow). For #SAT, the authors use BPNN and ApproxMC3 (among others) as baselines and show superior RMSE wrt. BPNN and better runtime wrt. ApproxMC3 on BIRD benchmark. A criticism to approximate neural #SAT solvers, which also applies to BPNN, is that unlike ApproxMC3 and other probably approximately correct (or PAC) counters, they do not provide any guarantees that the results fall within a confidence interval of the true model count. This makes runtime comparison to ApproxMC3 unfair and limits the use case of these types of solvers, unlike [2] which improved the heuristic of an exact solver. Having said that NSNet clearly beats the BPNN on these benchmarks which is an interesting result.


[1]: Li et al., “Combinatorial optimization with graph convolutional networks and guided tree search”

[2]: Vaezipoor et al. “Learning Branching Heuristic for Propositional Model Counting”

---

> ### Author Response · Authors · 2022-08-02
> **Response to Reviewer Tesa**
>
>
> Thank you for the thoughtful and in-depth comments. Below, we hope to address your stated points of our paper.
>
> ### Response to the questions
>
> > Comparisons between NSNet, NeuroSAT, and BPNN
>
> - NSNet is a novel GNN architecture that subsumes BP in the latent space and can be configured to solve both SAT and #SAT problems by performing marginal inference and partition function estimation respectively. It uses a novel graph representation of CNF formulas and performs message passing on _edges_ with the same aggregation operator as BP. Such design also enforces the permutation invariance and the negation equivariance of CNF formulas.
>
> - NeuroSAT uses the literal-clause graph representation of CNF formulas and uses a similar gated graph neural network (GGNN) that performs message passing on nodes. It can construct a possible assignment by decoding the literal embeddings for a SAT problem. However, it cannot compute the factor beliefs like BP, therefore, learning the Bethe approximation for #SAT is not applicable for the NeuroSAT architecture.
>
> - BPNN maintains the BP’s message passing procedure (not in the latent space) and only modifies factor-to-variable messages using a neural network. It takes the differences of the factor-to-variable messages in two iterations as input and outputs new differences. It is designed to estimate the partition function of general graphical models (not only for the #SAT) but fails to satisfy the negation equivariance of CNF formulas. Besides, it requires a predefined maximum number of variables in the factor to determine the dimension of its neural networks. Instead, NSNet is designed for SAT and #SAT problems and doesn’t have the constraint above.
>
> Given the difference between NSNet and other baselines, we believe the better performance of NSNet is due to the following reasons:
>
> - Compared with NeuroSAT, NSNet explicitly takes the probabilistic formulation of the SAT problem and captures BP in the GNN design to perform marginal inference, while NeuroSAT architecture is designed to directly construct a specific satisfying assignment.
>
> - Compared with BPNN, NSNet generalizes BP in the latent space rather than simply using neural networks to improve some scalars in BP, which is more flexible to learn a better inference algorithm. It also enforces some important properties of CNF formulas, which makes it more suitable for #SAT.
>
> > Number of parameters of NSNet, NeuroSAT, and BPNN
>
> In our paper, we set the feature dimension of all neural networks to 64 and layers of MLP to 3 to ensure a fair comparison. The parameters of these 3 neural networks are shown in the table below:
>
> | model | #parameters |
> |----------|-------------|
> | NSNet | 58,434 |
> | NeuroSAT | 116,673 |
> | BPNN | 599,171 |
>
> Although NSNet has the least number of parameters to learn, the novel GNN architecture makes it achieve better performance than NeuroSAT and BPNN on SAT and #SAT respectively.
>
> > Number of inference time message passing iterations of NSNet, NeuroSAT, and BPNN
>
> We studied more iterations of message passing during testing and report the best results for the neuro solvers. For SAT, in our evaluation of the SATLIB benchmark, the solving accuracy of NeuroSAT and NSNet increases when the number of iterations increases. As shown in Table 1 in our paper, the performance of NeuroSAT hits saturation after 500 iterations while NSNet only takes 50 iterations to converge. For #SAT, in our evaluation of the BIRD benchmark, performing 10 message passing iterations is enough for both NSNet and BPNN.

---

> > ### Author Response · Authors · 2022-08-02
> > **Response to Reviewer Tesa (cont.)**
> >
> > ### Clarification of the weaknesses
> >
> > > Novelty of the use of BP in the context of SAT (and #SAT)
> >
> > To the best of our knowledge, the use of BP has not been explored in a neural fashion for SAT, although BP has been explored for #SAT in a neural fashion. The existing neural versions of BP mainly replace some components (i.e. scalars) of BP with the neural networks (e.g. learning the damping ratio for the updates of factor to variable messages). In contrast, NSNet is designed to subsume BP in the latent space (i.e. each message is a high dimensional vector) by using a novel GNN architecture. Additionally, NSNet enforces the permutation invariance and the negation equivariance of CNF formulas, which other neural versions of BP (e.g. BPNN) fail to satisfy.
> >
> > > Whether neural SAT solvers perform a similar type of computations as NSNet
> >
> > We acknowledge neural messages themselves are not obviously interpretable, but we believe there are some important differences regarding the type of computations between NeuroSAT-like solvers and NSNet. Two factors contribute to such differences.
> >
> > -   First, the learning objective is very different. Existing neural SAT solvers like NeuroSAT directly solve a SAT problem by predicting (or constructing) a specific satisfying assignment, while NSNet performs marginal inference over all satisfying solutions. That is, by design, NSNet does not solve the SAT problem directly. Our experiments show that the former learning objective is less effective (although seemingly straightforward) and leads to quite low solving accuracy (results in Section 4.1). Without any local search, NSNet can achieve higher solving accuracy by simply rounding marginals.
> >
> > -   Second, the graph representation and the way of performing message passing are different. Existing neural SAT solvers use literal-clause graph representation and perform message passing among nodes, and such design cannot compute factor beliefs like BP. In contrast, NSNet uses a BP-like graph representation and performs message passing among edges.
> >
> > > Training NSNet needs a prior that the instances are satisfiable
> >
> > Our probabilistic formulation and unsupervised loss for SAT are based on the assumption that the given formulas are satisfiable. Indeed, we conjecture that all the GNN frameworks can only “solve” the satisfiable instances rather than unsatisfiable ones (predicting unsatisfiability without proof is relatively trivial, which we don’t consider as “solving”). For satisfiable instances, existing GNNs may help to find a satisfying assignment. But for unsatisfiable ones, they need to prove that _all_ of the possible assignments cannot satisfy the formula or construct an unsat proof, which is hard for GNN architectures to learn by performing message passing. The paper [1] gives a similar conclusion.
> >
> > [1] Chen, Ziliang, and Zhanfu Yang. "Graph neural reasoning may fail in certifying boolean unsatisfiability." _arXiv preprint arXiv:1909.11588_ (2019).
> >
> > > The novelty of obtaining a satisfying assignment from the variable bias and initializing an SLS solver
> >
> > Existing neural SAT solvers focus on predicting a satisfying assignment directly. In contrast, by design, NSNet is trained to estimate variable marginals among all satisfying assignments and needs a post-processing step to obtain a satisfying assignment. We show that simply rounding the estimated marginals by NSNet (without local search) is already more effective than directly predicting a satisfying solution like NeuroSAT. As stated in the appendix, with the estimated marginals, it is also flexible and straightforward to combine NSNet with many methods (CDCL solvers, SLS solvers, or the decimation algorithm) to construct a satisfying assignment. In our paper, we choose to integrate NSNet with a SLS solver because we find it efficient and introduce limited overhead.
> >
> > > Results of NSNet-Sparrow and BP-Sparrow on SAT problem
> >
> > For SAT, the performance of BP-sparrow and NSNet-sparrow is relatively close because the local search itself contributes to solving a large portion of the instances. Although NSNet gives a better initialization than BP (i.e. better solving accuracy of the initial assignments), the performance of a SLS solver may still depend on the local search procedure.
> >
> > > Guarantees of NSNet on #SAT problem
> >
> > For BP, it is guaranteed to return the exact model count if the graphical model of an instance is a tree. In the case the graph has cycles, the fixed point of BP is only guaranteed to be a local minimum of the Bethe free energy. By learning the Bethe approximation of the partition function, NSNet is likely to inherit some theoretical guarantee of BP. Although it is not yet clear whether our neural generalization of BP can provide the PAC-style guarantee like ApproxMC3, we empirically find that NSNet can provide quite good estimations while being orders of magnitude faster after training, motivating future rigorous theoretical analysis in this direction.

---

### Author Response · Authors · 2022-08-05
**Happy to answer further questions**

We hope our responses help to clarify some confusion/concerns about our paper. We are happy to answer further questions and/or elaborate on our contributions and evaluations.

---

### Meta-Review · Area_Chair_qvdx · 2022-08-27

**Recommendation:** Accept
**Confidence:** Less certain

**Metareview:**

This is a controversial paper. My thinking, however, is to largely agree with bWLw that the performance warrants publication even if that performance is largely due to details rather than major architectural innovation. I think that the overlap with Savari and Bortolussi is not a problem given the timing of the publication. The much smaller number of parameters compared to NeuroSAT and BPNN (see the response to Tesa) also seems significant.

**Award:**

No

---

### Decision · Program_Chairs · 2022-09-14

Accept